# THE LOSS LANDSCAPE OF OVERPARAMETERIZED NEURAL NETWORKS

## ABSTRACT

We explore some mathematical features of the loss landscape of overparameterized neural networks. A priori one might imagine that the loss function looks like a typical function from $\mathbb{R}^n$ to $\mathbb{R}$ - in particular, nonconvex, with discrete global minima. In this paper, we prove that in at least one important way, the loss function of an overparameterized neural network does not look like a typical function. If a neural net has $n$ parameters and is trained on $d$ data points, with $n > d$, we show that the locus $M$ of global minima of $L$ is usually not discrete, but rather an $n - d$ dimensional submanifold of $\mathbb{R}^n$. In practice, neural nets commonly have orders of magnitude more parameters than data points, so this observation implies that $M$ is typically a very high-dimensional subset of $\mathbb{R}^n$.

## 1 INTRODUCTION

In recent years, it has become clear that neural nets are incredibly effective at a wide variety of tasks. Why they work so well is less understood. During training neural nets implement a form of curve fitting, by minimizing a loss function $L : \mathbb{R}^n \to \mathbb{R}$. Evidence suggests that they work better when there are more parameters than data points. Here we explore the geometry of the loss landscape of overparameterized neural networks, as a first step toward understanding their unreasonable effectiveness.

A priori one might imagine that the loss function $L$ of a neural network looks like a typical function from $\mathbb{R}^n$ to $\mathbb{R}$ - in particular, nonconvex, with discrete global minima, many "bad" local minima that are not global minima, and complicated geometry. However, it turns out that in at least one important way, the loss function of an overparameterized neural network does not look like a typical function. In the overparameterized setting where the neural net has $n$ parameters and is trained on $d$ data points, with $n > d$, it is generally the case that $L$ is nonconvex. However, the locus $M \subset \mathbb{R}^n$ of global minima of $L$ is often not discrete. Indeed we find $M$ is often quite the opposite – rather than being a discrete set, $M$ is often an $n - d$ dimensional submanifold of $\mathbb{R}^n$. We expect this to be true in a very general setting. The neural network can be any depth, and while our strongest results are for feedforward neural networks, many of the results in this paper apply for a broader set of architectures. In practice, neural nets commonly have orders of magnitude more weights than data points, so in practical applications $M$ is typically a very high-dimensional submanifold of $\mathbb{R}^n$.

Our contributions include:

- Giving a heuristic that explains that in the overparameterized setting, generically the locus of global minima of the loss function should be an $n - d$ dimensional submanifold of $\mathbb{R}^n$, and that at a global minimum, the Hessian of $L$ has $d$ positive eigenvalues, $n - d$ eigenvalues equal to 0, and no negative eigenvalues.
- Proving that for a broad class of activation functions, a large enough feedforward neural network with one hidden layer can fit a fixed training dataset exactly.
- Proving that for an overparameterized feedforward neural network with a rectified smooth activation function, the locus of global minima for the loss function is a nonempty smooth $n - d$ dimensional submanifold of $\mathbb{R}^n$.

These results provide a theoretical setting for several lines of recent work. It has been widely suggested that gradient descent is successful in producing good solutions in deep learning because many

local minima are close to global minima and give similar results LeCun et al. (2015), Choromanska et al. (2015). As a result, the loss landscape has been studied in papers such as Freeman & Bruna (2017). Several authors have proved that under various strong assumptions, all local minima are global Liang et al. (2018), Ge et al. (2017), Laurent & Brecht (2017), Kawaguchi (2016). This is not true in "real world" settings for deep learning. But recent work has suggested that in some settings there are paths between local minima such that the value of the loss function stays low along the paths Garipov et al. (2018), Draxler et al. (2018). We show a stronger result in a more general setting. We prove that in a very general setting, there are not only one-dimensional paths between minima, but very high dimensional manifolds.

In Section 2, we study the geometry of the loss landscape in the general overparameterized setting. In Section 2.1 we study the global geometry of the landscape. We study the locus $M = L^{-1}(0)$ because $L$ is a nonnegative function, so as long as $M$ is nonempty, $M$ is the locus of global minima of $L$. We show that if nonempty, $M$ is an $n - d$ dimensional submanifold of $\mathbb{R}^n$. In Section 2.2, we consider the local geometry of the loss landscape near $M$, and show that if $M$ is nonempty then at a global minimum $m \in M$, the Hessian of $L$ has $n - d$ zero eigenvalues, $d$ positive ones, and no negative ones. This is consistent with previous observations in papers such as Chaudhari et al. (2016) and Wu et al. (2017).

In Section 3, we write down a proof of a commonly believed fact, that given a fixed dataset, a large enough one-hidden-layer feedforward neural net can memorize it exactly, i.e. learn it with zero training error. This is commonly assumed, and in the case that the activation function used is ReLU, proved in Zhang et al. (2016). We require a somewhat more general statement, but our argument is based on the one given there.

In Section 4, we combine the results of the previous two sections to show that in the setting of overparameterized feedforward neural networks, if they are wide enough then the locus $M = L^{-1}(0)$ is in fact a *nonempty* smooth $n - d$ dimensional submanifold of $\mathbb{R}^n$.

## 1.1 ASSUMPTIONS

In this paper, we will always consider the overparameterized setting. So in all of our analyses, we assume that the number of parameters $n$ of the neural net is greater than the number of data points $d$ that it is training on.

## 2 GEOMETRY OF THE LOSS LANDSCAPE $L$ - GENERAL CASE

### 2.1 POSITIVE DIMENSIONALITY OF $M$

Suppose we have a neural net of any architecture (e.g. feedforward, LSTM, etc.), with weights $(w_1, ...)$ and biases $(b_1, ...)$, $n$ parameters in total. Suppose this net uses a smooth activation function $\sigma$ and is training on a data set $\{(x_i \to y_i)\}$ with $d$ data points. Each entry in the data set is given as a pair of vectors, $x_i \in \mathbb{R}^p$ (e.g. a vector of pixel values) and $y_i \in \mathbb{R}$. We assume that the $x_i$ are distinct and that $n > d$.

Let $f_{w,b}$ be the function given by the neural net with the parameters $w, b$. For each data points $(x_i \to y_i)$, let $f_i(w, b) = f_{w,b}(x_i) - y_i$. Assume that each $f_i(w, b)$ is smooth in $w$ and $b$. For most choices of architecture, this is implied by the assumption that the activation function $\sigma$ is smooth. For example, for any feedforward neural network, if $\sigma$ is smooth then $f_i(w, b)$ is smooth for each $i$.

Let the loss function used in training be the commonly used

$$L(w, b) = \sum (f_{w,b}(x_i) - y_i)^2.$$

Define $f_i(w, b) : \mathbb{R}^n \to \mathbb{R}$ as

$$f_i(w, b) = f_{w,b}(x_i) - y_i,$$

so $L(w, b)$ can be written as

$$L(w, b) = \sum f_i(w, b)^2.$$

It is clear from the definition that $L(w, b) \geq 0$, so if nonempty, $M = L^{-1}(0)$ is the locus of global minima of $L$. Therefore, in this section we will focus on understanding the geometry of $M$ in the case that it is nonempty.

Note that

$$M = \bigcap M_i,$$

where

$$M_i = f_i(w, b)^{-1}(0).$$

The rest of this section is devoted to the following theorem.

**Theorem 2.1.** *In the setting described above, the set $M = L^{-1}(0)$ is generically (that is, possibly after an arbitrarily small change to the data set) a smooth $n - d$ dimensional submanifold (possibly empty) of $\mathbb{R}^n$.*

*Remark* 2.2. The loss function

$$\hat{L}(w, b) = \sum |f_i(w, b)|$$

is also regularly used. Note that

$$\hat{L}^{-1}(0) = L^{-1}(0),$$

so Theorem 2.1 also shows that the locus $\hat{M} = \hat{L}^{-1}(0)$ is a smooth $n - d$ dimensional submanifold of $\mathbb{R}^n$.

We start with a heuristic argument for Theorem 2.1.

**Heuristic:** For each $f_i$, the regular values of $f_i : \mathbb{R}^n \to \mathbb{R}$ are dense. So generically we expect that each $M_i = f_i^{-1}(0)$ is a smooth codimension 1 submanifold of $\mathbb{R}^n$. Generically the intersection of $d$ smooth codimension 1 submanifolds of $\mathbb{R}^n$ is a smooth codimension $d$ submanifold, so one would expect that $M = \bigcap M_i$ is smooth of codimension $d$.

With this heuristic in mind, we proceed to a proof of Theorem 2.1.

*Proof.* We construct a function $H$ related to $L$. Let $H : \mathbb{R}^n \to \mathbb{R}^d$ be defined as the function

$$H(w, b) = (f_1(w, b), ..., f_d(w, b)).$$

By construction, $L = |H|^2$, and importantly,

$$M = L^{-1}(0) = H^{-1}\left((0, ..., 0)\right).$$

If $(0, ..., 0)$ is a regular value of $H$, proceed to next paragraph. If not, pick any $\epsilon > 0$. By Sard's theorem, regular values of $H$ are generic. We may therefore choose a regular value $r = (r_1, ..., r_d)$ with $|r| < \epsilon$. We use $r$ to perturb the data, replacing $x_i \to y_i$ by $x_i \to \tilde{y}_i$ where $\tilde{y}_i = y_i + r_i$. Let

$$\tilde{f}_i(w, b) = f_i(w, b) - r_i = f_{w,b}(x_i) - \tilde{y}_i,$$
$$\tilde{L}(w, b) = \sum \tilde{f}_i^2,$$
$$\text{and } \tilde{H}(w, b) = (\tilde{f}_1(w, b), ..., \tilde{f}_d(w, b))$$
$$= H(w, b) - (r_1, ..., r_d).$$

Note that $|(x, y) - (x, \tilde{y})| < \epsilon$, meaning that after taking an arbitrarily small perturbation of the data, $(0, ..., 0)$ is a regular value of $\tilde{H}$.

After possibly replacing $H$ by $\tilde{H}$, $(0, ..., 0)$ is a regular value of $H$, meaning

$$M = H^{-1}((0, ..., 0))$$

is either empty or smooth of codimension $d$ in $\mathbb{R}^n$, as desired.

$\square$

## 2.2 THE LOCAL GEOMETRY OF THE LOSS FUNCTION NEAR $M$

**Proposition 2.3.** *Consider the submanifold $M = L^{-1}(0) = \bigcap M_i$, where $M_i = f_i^{-1}(0)$. If each $M_i$ is a smooth codimension 1 submanifold of $\mathbb{R}^n$, $M$ is nonempty, and the $M_i$ intersect transversally at every point of $M$, then at any point $m \in M$, the Hessian of $L$ evaluated at $m$ has $d$ positive eigenvalues and $n - d$ eigenvalues equal to 0.*

*Proof.* Let $\mathcal{H}(L)$ denote the Hessian of $L$. At every point $p \in \mathbb{R}^n$, the Hessian of $L$ evaluated at $p$, $\mathcal{H}(L)|_p$, is a real symmetric matrix, hence has a basis of eigenvectors. Since $m$ is a minimum (in fact a global minimum) of $L$, the eigenvalues of $\mathcal{H}(L)|_m$ are nonnegative. Since $M$ is $n - d$ dimensional, the kernel of $\mathcal{H}(L)|_m$ is at least $n - d$ dimensional. It remains to show that it is also at most $n - d$ dimensional.

Well, $L = \sum_{i=1}^{d} f_i^2$, so

$$\mathcal{H}(L) = \sum_{i=1}^{d} \mathcal{H}(f_i^2).$$

Let us consider each summand. The linear transformation $\mathcal{H}(f_i^2)$ has $n - 1$ eigenvalues equal to 0 and a unique nonzero eigenvalue $\lambda_i$. This follows from the fact that $M_i$ is a smooth codimension 1 submanifold. Furthermore, $\lambda_i$ is positive as 0 is a global minimum of $f_i$. Let $v_i$ denote the eigenvector corresponding to this positive eigenvalue.

The vectors $v_1, ..., v_d$ are linearly independent because the $M_i$ intersect transversally at $m$. An elementary linear algebra calculation shows that if an $n \times n$ matrix $A$ is the sum of $d$ matrices $A = \sum_{i=1}^{d} A_i$, $n > d$, where each $A_i$ has a unique nonzero eigenvalue $\lambda_i$ and the corresponding vectors $v_i$ are all linearly independent, then $A$ has $d$ nonnegative eigenvalues and $n - d$ eigenvalues equal to 0. We conclude that in our setting, $\mathcal{H}(L)$ has $d$ positive eigenvalues and $n - d$ eigenvalues equal to 0, as desired.

$\square$

## 3 FEEDFORWARD NEURAL NETS CAN FIT POINTS

We have seen that in a very general setting, assuming almost nothing about the architecture of a neural network, the locus $M = L^{-1}(0)$ is, if nonempty, a positive dimensional submanifold of $\mathbb{R}^n$. However, whether $M$ is nonempty depends on the chosen architecture and other details of implementation. In this section, we address the issue of nonemptiness of $M$ in a (still fairly general) setting of feedforward neural networks with nice activation functions.

We start by recalling the definition of a feedforward network, and setting some notation. A feedforward neural network is specified by the data of a directed acyclic graph $G(V, E)$, a parameter vector $p$ in $\mathbb{R}^m$, a labeling $\pi : E \cup V \to \{1, ..., m\}$, and an activation function $\sigma$. Let $V_{in}$ denote the set of input vertices (the vertices in V with no incoming edges) and $V_{out}$ the set of output vertices (those with no outgoing edges).

The labeling $\pi$ associates to each edge or vertex a parameter. The parameters for the edges are commonly referred to as weights, and the parameters for the vertices are called biases. We often reflect this by referring to a pair of vectors $w$ of weights and $b$ of biases instead of a single parameter vector $p$.

Given a fixed parameter vector $p$, (or the pair $(w, b)$), we can construct a function $f_{G,\pi,\sigma,p} : \mathbb{R}^{|V_{in}|} \to \mathbb{R}^{|V_{out}|}$ as follows. For any input node $v \in V_{in}$, its output $o_v$ is the corresponding coordinate of the input vector $x \in \mathbb{R}^{V_{in}}$. For each internal node $v$, its output is defined recursively as

$$o_v = \sum_{u \to v \in E} \sigma(p_{\pi(u \to v)} \cdot o_u + p_{\pi(v)}),$$

or perhaps more familiarly with weights and biases,

$$o_v = \sum_{u \to v \in E} \sigma(w_{\pi(u \to v)} \cdot o_u + b_{\pi(v)}).$$

For output nodes $v \in V_{out}$, no non-linearity is applied and their output

$$o_v = \sum_{u \to v \in E} w_{\pi(u \to v)} \cdot o_u + b_v$$

determines the corresponding coordinate of the computed function $f_{G,\pi,\sigma,p}$.

We denote the hypothesis class of functions computable by the neural net using some choice of parameters by $\mathcal{N}(G, \pi, \sigma) = \{f_{G,\pi,\sigma,p} | p \in \mathbb{R}^m\}$.

The term neural network can refer to a specific function, the hypothesis class of functions, or other related objects. When we use the term neural network, we generally mean a hypothesis class of functions, or this class of functions along with a framework for choosing parameters $p$ such that the resulting function $f_{G,\pi,\sigma,p}$ fits the training data reasonably well.

When we talk about neural networks, we generally consider $G$, $\pi$, and $\sigma$ fixed, and the task is to find a good choice of parameters $p$. We often suppress the fixed objects in our notation. For example, in this paper we will commonly use the notation $f_{w,b}$ to denote the function computed by a neural net with a choice of parameters $w$ weights and $b$ biases, and implicitly $G$, $\pi$, and $\sigma$ are fixed in the discussion.

## 3.1 RECTIFIED ACTIVATION FUNCTIONS

**Definition 3.1.** *We call a continuous function $\sigma : \mathbb{R} \to \mathbb{R}$ rectified if*

$$\sigma(x) = \begin{cases} 0, & x \le 0, \\ \text{monotonic increasing}, & x > 0. \end{cases}$$

Our strongest results apply to feedforward neural networks whose activation function $\sigma$ is rectified smooth.

The set of rectified smooth functions doesn't contain most commonly considered activation functions. For example, ReLU, tanh, and sigmoid are not rectified smooth functions. However, ReLU modified by smoothing the corner at 0, commonly used in practice when implementing neural networks, is. Translated and truncated tanh and sigmoid are as well, and in practice behave similarly to tanh and sigmoid in neural networks.

For a concrete definition of a smooth rectified activation function, we make the following definition.

**Definition 3.2.** *Let $smooLU$ be defined as*

$$smooLU(x) = \begin{cases} 0, & x \le 0, \\ x \exp(-1/x), & x > 0. \end{cases}$$

In the remainder of this paper, the reader can take the activation function $\sigma$ to be smooLU if desired. The function smooLU is similar to softplus, which is commonly used in neural nets.

With these concepts in hand, we proceed to the main result of this section. For the next lemma we don't need to assume $\sigma$ is smooth.

**Lemma 3.3.** *Fix a data set $S = \{(x_i, y_i)\}$ with $d$ data points, $x_i \in \mathbb{R}^p$, $y_i \in \mathbb{R}$, and a rectified activation function $\sigma$. Assume that the vectors $x_i$ are distinct, i.e. no two are equal. For any $h \ge d$, there exists a feedforward neural network $f_{w,b}$ with 1 hidden layer of width $h$ and activation function $\sigma$ that represents $S$ with zero training error. In fact, we produce weights $w, b$ such that $f_{w,b}(x_i) = y_i$ for all $i$. Here the number of parameters $n$ governing the family of neural nets we consider in the construction is $n = 2d + p$.*

*Proof.* A feedforward neural net of this form is a function

$$f_{w,b}(x) = M_2\sigma(M_1 x - b_1) - b_2,$$

where $M_1, M_2$ are linear transformations

$$M_1 : \mathbb{R}^p \to \mathbb{R}^h \text{ and } M_2 : \mathbb{R}^h \to \mathbb{R},$$

and $b_1, b_2$ are vectors

$$b_1 \in \mathbb{R}^h \text{ and } b_2 \in \mathbb{R}.$$

The weights $w$ are the entries of $M_1$ and $M_2$, and the biases $b$ are the entries of $b_1$ and $b_2$.

We use the convention that $\sigma$ applied to a vector denotes component-wise evaluation:

$$\sigma(v_1, ..., v_m) = (\sigma(v_1), ..., \sigma(v_m)).$$

In our construction, we will take $b_2 = 0$ and all the rows of $M_1$ to be equal to a single vector $a = (a_1, ..., a_p)$. We also take $h = d$ the number of data points. If $h > d$, one can set all the weights and biases of the nodes after the first $d$ to be 0, so it suffices to make the construction for $h = d$. So we are looking for a function $f_{w,b}$ of the form

$$f_{w,b}(x) = \sum_{j=1}^{d} m_j \sigma(a \cdot x - b_j)$$

such that $f_{w,b}(x_i) = y_i$ for all $i$. Our job is to find values for the $2d + p$ parameters $m_1, ..., m_d$, $a_1, ..., a_p, b_1, ..., b_d$ satisfying these conditions.

First, we choose a vector $a \in \mathbb{R}^p$ such that $\{a \cdot x_i\}$ are distinct. This is possible because we assumed the $x_i$ are distinct and a general projection from $\mathbb{R}^p \to \mathbb{R}$ will preserve that property. Up to a reordering of the points $x_i$, we can assume that $\{ax_i\}$ are increasing. Choose $x_0$ such that $ax_0 = ax_1 - 1$. Let

$$b_i = \frac{ax_{i-1} + ax_i}{2}.$$

Now, to arrange that $f_{w,b}(x_i) = y_i$ we require

$$y_i = \sum_j \sigma(a \cdot x_i - b_j) m_j. \tag{3.1}$$

We express Equation 3.1 using the notation of matrix multiplication,

$$\begin{pmatrix} y_1 \\ \vdots \\ y_d \end{pmatrix} = \begin{pmatrix} \sigma(a \cdot x_1 - b_1) & \cdots & \sigma(a \cdot x_1 - b_d) \\ \vdots & \ddots & \vdots \\ \sigma(a \cdot x_d - b_1) & \cdots & \sigma(a \cdot x_d - b_d) \end{pmatrix} \begin{pmatrix} m_1 \\ \vdots \\ m_d \end{pmatrix}. \tag{3.2}$$

Let us denote the matrix whose $i, j^{th}$ entry is $(\sigma(a \cdot x_i - b_j))$ by $A$.

By construction, $A$ is lower triangular with nonzero entries on the diagonal, hence invertible. So we set the weights by

$$\begin{pmatrix} m_1 \\ \vdots \\ m_d \end{pmatrix} = A^{-1} \begin{pmatrix} y_1 \\ \vdots \\ y_d \end{pmatrix}.$$

This gives us a choice of weights and biases for a neural network with one hidden layer which has learned the data $\{(x_i, y_i)\}$ with zero error.

$\square$

Since a feedforward neural network with one hidden layer can always be embedded in a deeper feedforward net with sufficient width, we have the following corollary of Lemma 3.3.

**Corollary 3.4.** *Fix a data set $S = \{(x_i, y_i)\}$ with $d$ data points, $x_i \in \mathbb{R}^p$, $y_i \in \mathbb{R}$, and a rectified activation function $\sigma$. Assume that the vectors $x_i$ are distinct, i.e. no two are equal. Among feedforward neural networks with activation function $\sigma$ and $T$ hidden layers, last hidden layer of width $h \geq d$, remaining hidden layers of arbitrary width, there exists a choice of weights so that the neural network $f_{w,b}$ represents $S$ with zero training error.*

*Proof.* We can adapt the construction in the proof of Lemma 3.3 to the case of a deeper net by simply choosing a subset of the nodes of the deeper net to use. Suppose the $T$ hidden layers have widths $h_1, ..., h_T$. The nodes we use are the first node of each of the first $T - 1$ hidden layers, and the first $d$ nodes of the last hidden layer. Let the first linear transformation $M_1 : \mathbb{R}^p \to \mathbb{R}^{h_1}$ be constructed as the vector $a$ as in Lemma 3.3 in the first row and 0 in all other rows. Choose $b_1$ so that $a \cdot x_i - b_1$ is positive for all $i$. In the remaining layers, let the matrix $M_j$, $2 \le j \le T$ be 1 in the top left entry and 0 in all the others. Note that the property that $a \cdot x_i - b_1$ are all positive and distinct is preserved through the layers, because the activation function $\sigma$ is rectified. Finally, let $M_{T+1}$ be constructed as in Lemma 3.3. □

## 4 Geometry of the loss landscape $L$ - feedforward case

### 4.1 Positive dimensionality of $M$

Suppose we have a feedforward neural net of any depth and last hidden layer of width $h > d$. Suppose this net uses a rectified smooth activation function $\sigma$ and is training on a data set $\{(x_i \to y_i)\}$ with $d$ data points. Each entry in the data set is given as a pair of vectors, $x_i \in \mathbb{R}^p$ (e.g. a vector of pixel values) and $y_i \in \mathbb{R}$. We assume that the $x_i$ are distinct.

Let $f_{w,b}$ be the function given by the neural net with the parameters $w, b$. As before, let the loss function used in training be the commonly used

$$L(w, b) = \sum (f_{w,b}(x_i) - y_i)^2.$$

**Theorem 4.1.** *In the setting described above, the global minimum of $L$ is generically (that is, possibly after an arbitrarily small change to the data set) equal to 0, and the set of global minima $M = L^{-1}(0)$ is a nonempty smooth $n - d$ dimensional submanifold of $\mathbb{R}^n$.*

*Proof.* By Theorem 2.1, $M = L^{-1}(0)$ is a smooth $n - d$ dimensional submanifold of $\mathbb{R}^n$. It remains only to show that $M$ is nonempty. But that is exactly the guarantee of Lemma 3.3, given that we have assumed that $\sigma$ is a rectified smooth activation function and that the architecture of the networks is a feedforward network whose last hidden layer has width at least $d$. □

*Remark 4.2.* The results in this paper can all be modified to accommodate the case that $y_i \in \mathbb{R}^\ell$ (e.g. a space of labels) where $\ell > 1$. The argument in Theorem 2.1 is modified by letting $H : \mathbb{R}^n \to \mathbb{R}^{\ell d}$ be $(f_1(w, b), ..., f_d(w, b))$ where $f_i(w, b)$ is now a function from $\mathbb{R}^n$ to $\mathbb{R}^\ell$, and we find that $M$ is codimension $\ell d$. The construction in Lemma 3.3 and Corollary 3.4 is modified by having $\ell$ copies of the neural network, one for each component of $y_i$. The parameter bounds change appropriately. With strengthened versions of those results, the argument in Theorem 4.1 extends without modification to the more general setting.

## 5 Discussion

In this paper, we note that far from being a typical function from $\mathbb{R}^n$ to $\mathbb{R}$, the loss function of an overparameterized neural network has some special geometric properties. This is compatible with several recent observations. For example, in Chaudhari et al. (2016), it was empirically observed that at solutions found by training neural networks with standard methods, the Hessian of the loss function tended to have many zero eigenvalues, some positive eigenvalues, and a small number of negative eigenvalues, which tended to be small in magnitude compared to the positive eigenvalues. Similarly, in Wu et al. (2017), Wu, Zhu, and E train some small deep neural networks and compute the spectrum of the Hessian at points obtained after training. They too observe that across several different models and datasets most of the eigenvalues are approximately zero, with few negative eigenvalues. In fact, they conjecture that "the large amount of zero eigenvalues might imply that the dimension of this manifold is large, and the eigenvectors of the zero eigenvalues span the tangent space of the attractor manifold. The eigenvectors of the other large eigenvalues correspond to the directions away from the attractor manifold." We have proved a more precise version of that statement here.

Note that Theorem 2.1 applies in a very general setting. In particular, it holds for a neural network of any architecture learning a data set $S = \{x_i, y_i\}$, as long as the loss function $L$ is of the form

$$L(w, b) = \sum_i |f_{i;w,b}(x_i) - y_i|^a, a \geq 1,$$

and each $f_{i;w,b} : \mathbb{R}^n \to \mathbb{R}$ is smooth. In practice, these conditions usually hold as long as a smooth activation function is used. (In practice, usually all the $f_{i;w,b}$ are even the same.) So in most settings, we are guaranteed that $M$ is an $n - d$ dimensional possibly empty submanifold of $\mathbb{R}^n$.

To determine whether $M$ is nonempty requires an argument specific to the chosen architecture and details of implementation, such as the one we gave here for feedforward neural networks with rectified smooth activation functions.

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
