# OpenReview forum: "The loss landscape of overparameterized neural networks"
_ICLR.cc/2019/Conference_

### Official Review · AnonReviewer2 · 2018-10-29
**A short and concise theoretical paper, but I find the contribution limited**

**Rating:** 5
**Confidence:** 4

**Review:**

The paper shows that the set of global minimums of an overparametrized network with smooth activation function is almost surely a high dimensional manifold. In particular, the dimension of this manifold is exactly n-d, where n is the number of parameters and d is the number of datas. To the best of my knowledge, this is the first proper proof of such non-surprising result.

The theoretical analysis is a straightforward combination of neural network's expressiveness result and some classical theorems in the field of differential topology. However, the assumption on the overparametrized neural network is somehow unrealistic since it requires that at least one layer has no less than d neurons, which is as many as the number of datas. This is usually not the case. Moreover, the result is to some extent "asymptotic", in the sense that a small perturbation in terms of data may be required in order to make the statement holds.

More importantly, the result does not provide any useful characterization distinguishing stationary point/local minimum versus global minimum. It might be possible that the set of stationary points is also a manifold with very high dimension, which is indeed supported by the argument listed in the bottom of page 7: "the Hessian of the loss function tended to have many zero eigenvalues, some positive eigenvalues, and a small number of negative eigenvalues". It is possibly the case that the set of stationary points has even higher dimension. This suggests that the dimensionality itself is not right indicator, but the difference in terms of dimension between different type of stationary points that matters.

Overall, the paper is short and concise but I find the contribution a bit limited.

---

> ### Author Response · Authors · 2018-11-18
> **Author response**
>
> We thank the reviewer for the helpful feedback.  This review brings up three main points, which we address here.
>
> 1.  We prove two theorems in this paper.  The first theorem holds very generally, and requires almost no assumptions about the architecture of the neural net.  Informally, one can think of this theorem as saying that one should generically expect the locus $M=L^{-1}(0)$ to be a manifold of dimension $n-d$.  The second theorem invokes several assumptions.  The purpose of these assumptions is simply to prove that $M$ is nonempty.  We expect that $M$ is nonempty in many cases beyond what we prove here, and leave it to future work.
>
> But one way to frame it is that as soon as your neural network, whatever it is, has the capacity to perfectly fit the training data, then the dimension of $M$ is $n-d$.  So if no layer has width $d$ but you know that your network has the capacity to perfectly fit the training data, the conclusion of the second theorem still holds.
>
> 2.  It is true that a small perturbation of the data may be necessary, but this perturbation can be chosen as small as you like.  To be precise, if you fix any $\epsilon > 0$, a perturbation such that each new label is within $\epsilon$ of the original label can be found for which the theorem holds.
>
> 3.  This is a good point.  We are not currently able to compute the dimension of the locus of local but non-global minima.  However, while it is logically possible for loci of local non-global minima to have large dimensions, even larger than the dimension of the locus of global minima $M$, we would not expect this to be the case.
>
> A generic function has isolated critical points.  For a positive dimensional critical manifold to appear requires something special to happen.  In this paper we show that something special indeed happens at the locus of global minima $M = L^{-1}$, and we prove that its dimension is positive.  However, our argument relies on the fact that we are looking at the preimage of 0.  It would not apply to any other minima.  We agree that it would be interesting to study the dimension of the loci of local non-global minima.

---

> > ### Comment · AnonReviewer2 · 2018-11-22
> > **Does the result on landscape hold for any smooth function?**
> >
> > I thank the authors for the clarification.
> >
> > I totally agree with the authors that the result of the paper holds as long as the network can perfectly fit the data. However, this is somehow a tautology in my opinion. Since given any n-dimensional smooth function f, assuming f can perfectly fit the data, we are able to obtain the same landscape result. (please correct me know if I am wrong)
> >
> > Then the core question becomes how to achieve this perfect fitting property in an efficient way. Naively, we can use any family of functions which satisfies the universal approximation property (like polynomials or Fourier series). With the polynomial interpolation, a high-dimensional polynomial can also fit perfectly the data points, so it enjoys the same (n-d) dimensional manifold of global minima. However, we might not be able to benefit from this fact to efficiently find the coefficients of the polynomials.
> >
> > In my opinion, the success of deep learning implicitly suggests there is something spectacular in the structure of neural network, which is not true for any universal approximation family. I think a more interesting/important question will be how to take account of the structure/architecture of the network into the landscape.

---

> > > ### Author Response · Authors · 2018-11-23
> > > **Reply**
> > >
> > > We thank the reviewer for their question.
> > >
> > > Can you clarify what you mean by an n-dimensional smooth function f?  We are not able to answer your question as stated, but would like to address your question.
> > >
> > > We agree that understanding why solving regression problems using neural nets works better than, for example, using polynomials or Fourier series would be very interesting.  We don't aim to address that question here.  For now our aim is to uncover some aspects of the geometry of neural networks, and in doing so, explain some empirical observations about the loss landscape of neural networks that have been discussed in several recent papers.

---

> > > > ### Comment · AnonReviewer2 · 2018-11-23
> > > > **Clarification**
> > > >
> > > > Let me rephrase the question with a concrete example in the following way: now instead of using neural networks,  replace the function f_{w,b} by a polynomial, where w, b are the coefficients or the polynomial. The number of coefficients is n. (overparametrization somehow equivalent to using a high order polynomial)
> > > >
> > > > As far as I understand, the following statement is still valid:
> > > > Assume the polynomial f can perfectly fit the data, then the dimension of $M$ is $n-d$.
> > > >
> > > > What I am trying to say is such theorem does not involve any property of neural network, it only depends on
> > > > 1) smoothness, which allows using differential topology
> > > > 2) being universal approximator
> > > > If assuming the function has the capacity to perfectly fit the training data, then 2) is no longer needed so the result only relies on the property of smoothness.

---

> > > > > ### Author Response · Authors · 2018-11-24
> > > > > **Reply**
> > > > >
> > > > > Yes, you are exactly correct.  The example of an n-dimensional space of polynomials is a great example of a setting in which Theorem 2.1 holds.  If one uses L2 loss, then it is correct that the locus M = L^{-1}(0) is a nonempty manifold of dimension $n-d$ as long as there exists a polynomial in the chosen space that can perfectly fit the data.
> > > > >
> > > > > As you say, the first half of this paper applies in a much more general setting than only neural networks.  In Theorem 2.1, we state precisely the assumptions under which the locus of global minima of the loss function L is a smooth manifold of dimension $n-d$.  All those assumptions hold in the case of an n-dimensional space of polynomials with L2 loss.
> > > > >
> > > > > One take home message of this paper is that several recent empirical observations can be explained by some properties of the geometry of the loss function which are very general, and hold in a more general setting than only neural networks.

---

### Official Review · AnonReviewer3 · 2018-11-02
**The results from the paper are sort of known in previous literature, yet the proof in the paper was still smart and innovative.**

**Rating:** 7
**Confidence:** 3

**Review:**

This paper gave an interesting theoretical result that the global minima of an overparameterized neural network is a high-dimensional sub-manifold. This result is particularly meaningful as it connected several previous observations about neural networks and a indirect evidence for why overparameterization for deep learning has been so helpful empirically.

The proof in the paper was smart and the rough logic was quite easy to follow. The minor issue is that the proof in the paper was too sketchy to be strict. For example, in proof for Thm 2.1, the original statement about Sard’s theorem was about the critical values, but the usage of this theorem in the proof was a little indirect. I can roughly see how the logic can go through, but I still hope the author can give more detailed explaining about this part to make the proof more readable and strict.

Overall, I think the result in this paper should be enough to justify a publication. However, there’re still limitations of the result here. For example, the result only explained about the fitting on training data but cannot explain at all why overfitting is not a concern here. It also didn’t explain why stochastic gradient descent can find these minima empirically. In particular, even though the minima manifold is n-d dimensional, it’s still a zero-measure set which will almost never get hit with a random initialization. Of course, these are harder questions to explore, but maybe worthy some discussion in the final revision.

---

> ### Author Response · Authors · 2018-11-18
> **Author response**
>
> We thank the reviewer for the helpful feedback.
>
> We'd be happy to provide more details in the proof of Theorem 2.1.  Would the following clarification help?  The statement of Sard's theorem on wikipedia is that under certain conditions, the critical values of a function $f$ are measure 0.  An equivalent and alternative formulation which we use in this paper is that under those conditions, the regular values of $f$ are full measure.
>
> The reviewer suggests several further directions, which we think are valuable.  We have also wondered about many of these, but don't at the time have answers to them.  We hope this paper will serve as a first step for exploring the more difficult questions the reviewer brings up.

---

### Official Review · AnonReviewer1 · 2018-11-03
**Interesting but contribution is minor and doesn't meet the standards. Would recommend rejection.**

**Rating:** 5
**Confidence:** 4

**Review:**

This paper considers over-parametrized neural networks (n > d), where n is the number of parameters and d is the number of data, in the regression setting. It consists of three main results:

a) (Theorem 2.1) If the output of the network is a smooth function of its parameters and global minimum with zero squared loss is attainable, then the locus of global minima is a smooth $n - d$ dimensional submanifold of R^n.
b) (Lemma 3.3) A neural network with a hidden layer as wide as the number of data points can attain global minima with zero loss.
c) (Theorem 4.1) If a neural network has a hidden layer as wide as the number of data points and is a smooth function of its parameters, then the locus of global minima with loss zero is a smooth $n - d$ dimensional submanifold. This is just a combination of a) and b).

I think the only contribution of this paper is Theorem 2.1. It is already well-known (e.g., Zhang et al. 16’) that a network with a hidden layer as wide as the number of data can easily attain zero empirical error. The authors claim that it is an extension over ReLU, but the extension is almost trivial. Theorem 4.1 is just a corollary from Theorem 2.1 and Lemma 3.3.

Theorem 2.1 is an interesting observation, and its implication that the Hessian at minima has only d positive eigenvalues is interesting and explains empirical observations.

From my understanding, however, the n - d dimensional submanifold argument can be applied to any regular values of H(w,b) (not necessarily global minima), so the theorem is basically saying there are a lot of equivalence classes in the parameter space that outputs exactly the same outcome. If my understanding is correct, this result does not give us any insight into why SGD can find global minima easily, because we can apply the same proof to spurious local minima and say, “look, the locus of this spurious local minimum is a high-dimensional submanifold, so it is easy to get stuck at poor local minima.”

Moreover, the notation used in this paper is not commonly used in deep learning community, thus might confuse the readers. $n$ is more commonly used for the number of data points and $d$ is used for some dimension. Also, there are overloading uses of $p$:
* In Section 2.1 it is used for the dimension of the input space
* In the proof of Prop 2.3 it is now a vector in R^n
* In section 3 it is now a parameter vector of a neural network
* Back to the input dimension in Lemma 3.3 and Section 4

Also, the formulation of neural networks in Section 3 is very non-standard. If you are to end up showing some result on standard fully-connected feed-forward networks, why not just use, for example, $W_2 \sigma(W_1 x_i + b_1) + b_2$? To me, this graph theoretical formulation only adds confusion. Also at the end of page 4, there is a typo: the activation function (\sigma) must be applied after the summation.

Overall, I believe this paper does not meet the standards of ICLR at the moment. I would recommend rejection.

---

> ### Author Response · Authors · 2018-11-18
> **Author response**
>
> We thank the reviewer for the helpful feedback.
>
> Thanks for suggesting some modifications of the letters used and catching a typo.  As to the definition of neural networks, we state a fairly general definition because that is the setting of Theorem 2.1.  We only restrict to the special case of a fully connected feedforward network in Theorem 2.3.
>
> Now we turn to the reviewer's central concern.  The proof that $M = L^{-1}(0)$ is a manifold of dimension $n-d$ depends in an important way on the fact that we are studying the pre-image of 0, and this argument would not apply to any other minima - not even global minima of $L$ if the value of $L$ at the global minimum is not 0.  It is essential to the proof that 0 has the property that the sum of $d$ nonnegative numbers is 0 if and only if each summand is 0.
>
> We would not characterize Theorem 2.1 as "saying there are a lot of equivalence classes in the parameter space that outputs exactly the same outcome".  It is true that there may be multiple parameters that name the same function, e.g. there are many symmetries if ReLU is used as the activation function.  However, the parameters in $M$ generally will not all be names for the same function - rather, they encode many different functions, all of which share the property that they pass through the given $d$ data points.
>
> As a simplest example, consider a parameter space of lines in $R^2$, y=ax+b.  If we ask for all the lines that pass through the point $(1,3)$, we can find many distinct lines (the graphs of distinct functions) that pass through the point.  They are different lines - what they have in common is that they all pass through the point $(1,3)$.

---

> > ### Comment · AnonReviewer1 · 2018-11-19
> > **Reply to the response**
> >
> > I appreciate the authors for the response.
> >
> > I still do not understand why “global minimum with zero” is so important in the proof of Theorem 2.1. It just says that (0,0,...,0) is a regular value of H (or \tilde H), so M is a smooth n-d dim submanifold (if not empty). What if I pick another regular value (let's say, (1,0,...,0))? What prevents me from applying the same argument to these regular values?

---

> > > ### Author Response · Authors · 2018-11-20
> > > **Reply**
> > >
> > > Thank you for the reply.  We are glad to have the opportunity to clear up this point.  There are two functions - the loss function L we are interested in, and an auxiliary function H we construct to help us understand the geometry of L.  L is a function from R^n to R, while H is a function from R^n to R^d.
> > >
> > > You are entirely correct that if one takes any regular value (r_1, ..., r_d) of H, the preimage of that point is a smooth codimension $n-d$ dimensional submanifold.  However, in general, H^{-1}(r_1, ... r_d) is not L^{-1} of anything.  The special property of 0 that we are using is that L^{-1}(0) exactly equals H^{-1}(0,...0).

---

> > > > ### Author Response · Authors · 2018-11-23
> > > > **Additional comment**
> > > >
> > > > We include a more extensive explanation, in case it helps to clarify things.  In this paper, our aim is to understand the locus of global minima of L, a subset of the critical locus of the function L.  A priori our goal has nothing to do with the regular values of any function.  In the course of our proof, we construct a function H (or if necessary, \tilde{H}) with the nice property that (0,...,0) is a regular value of H and that H^{-1}(0,...,0) is equal to L^{-1}(0).  So we replace the more difficult problem of understanding a critical locus of $L$ with the more tractable problem of understanding the preimage of a regular value of H, aka a level set of H.
> > > >
> > > > In general studying the level sets of L is a very different problem.  For example, for most (a subset of R of full measure) values a, a is a regular value of L and L^{-1}(a) is a smooth n-1 dimensional manifold.  And when a is not a regular value of L, it is possible that some points of L^{-1}(a) are critical points of L and that at other points of L^{-1}(a), the map L is smooth.
> > > >
> > > > So the locus M=L^{-1}(0) we study in this paper has many special properties, one of which is that every point in M is a critical point of L.  We do not know of any reason to expect that for any other value a, L^{-1}(a) also has that property.

---

> > > > > ### Comment · AnonReviewer1 · 2018-11-27
> > > > > **Reply**
> > > > >
> > > > > I appreciate the authors for the clarification. I get your point; the argument can be applied to any regular value (r_1, ..., r_d) of H, but the points in M are not necessarily critical points of L. In contrast, for (0, ..., 0), the points in M are all global minima of L. I admit that I made a false statement about "local minima" in the review.
> > > > >
> > > > > However, even with this clarification, I think the contribution of this paper is rather limited, in the sense that:
> > > > > 1) The result doesn't always work; it requires additional noise injection when (0, ..., 0) is not a regular value of H.
> > > > > 2) As pointed out by Reviewer2, the theorem doesn't provide comparison between global minima and other critical points of L.
> > > > > 3) After Section 2, the remaining theorems do not make significant contribution, in my opinion.
> > > > >
> > > > > I'll adjust my rating accordingly.

---

> > > > > > ### Author Response · Authors · 2018-12-06
> > > > > > **Response**
> > > > > >
> > > > > > Great.  Glad we reached a common understanding about the special properties of global minima that were used in the argument.  Thanks for helping us clarify that.
> > > > > >
> > > > > > We wanted to make a brief clarification about your first concern.  It is true that given a set of data points {(x_i,y_i)}, x_i in R^a, y_i in R^b, $M$ may not be a smooth n-d dimensional manifold.  However, the set of data points in R^{a X b X d} for which this fails has measure 0.  So the probability that your data points have this property is 0.
> > > > > >
> > > > > > Furthermore, if your data set is one for which $M$ is not a smooth n-d dimensional manifold, you can choose any arbitrarily small \epsilon > 0 and there exists a perturbation of the data set so the distance between each new (x_i', y_i') and (x_i, y_i) is smaller than \epsilon.  So in real world applications, this should not be an issue.

---

### Public Comment · (anonymous) · 2018-11-06
**The result is based on heuristic, hence an unrigorous work.**

The paper aims to characterize the loss landscape of over-parameterized neural networks, which fit a fixed training dataset exactly. The authors try to use the regular value theorem to show the set of global minima is a smooth $n - d$ dimensional submanifold of $R^n$, where $n$ is the number of parameters of a neural network and $d$ is the number of data. The proof of this claim is only heuristic, namely, by Sard's theorem, the set of critical values of a smooth function between two smooth manifolds has Lebesgue measure zero. Unfortunately, this is insufficient to assert that global minimum zero, if exists, is a regular value. There is a similar attempt to study this problem at CVPR 2018 (check the link below), which shows that global minimum zero is not a regular value of the loss function with over-parameterized neural networks.

http://openaccess.thecvf.com/content_cvpr_2018/html/Shen_Towards_a_Mathematical_CVPR_2018_paper.html

---

> ### Author Response · Authors · 2018-11-18
> **Author response**
>
> We thank the commenter for this comment.
>
> We'd like to clarify that after the statement of Theorem 2.1, we give a heuristic for why one might expect this theorem to be true.  We then give a formal proof of the theorem.  We do not claim that 0 is a regular value of $L$.  Rather, the proof establishes that after perhaps an arbitrarily small perturbation of the data set, 0 is a regular value of each $f_i$.
>
> Thank you for bringing our attention to the paper mentioned.  We did not see it stated in that paper that "global minimum zero is not a regular value of the loss function with over-parameterized neural networks", but in any case, this paper does not contradict that statement.  Namely, in this paper, we do not prove that 0 is a regular value of the loss function $L$.

---

### Comment · Area_Chair1 · 2018-12-06
**quick question**

I am wondering whether the results imply that all the global minima form a *connected* n-d dimensional manifold?

---

> ### Author Response · Authors · 2018-12-06
> **Response**
>
> That is an excellent question, thanks for bringing it up.  No, they do not.  In fact, under the assumptions used in this paper, you can find examples where $M$ is not connected.
>
> That being said, we would like to know under what assumptions $M$ is connected.  We don't currently have a theorem about that.

---

### Meta-Review · Area_Chair1 · 2018-12-08

**Confidence:** 5
**Recommendation:** Reject

**Metareview:**

The paper proves that the locus of the global minima of an over-parameterized neural nets objective forms a low-dimensional manifold. The reviewers and AC note the following potential weaknesses:
--- it's not clear why the proved result is significant: it neither implies the SGD can find a global minimum, nor that the found solution can generalize. (Very likely, most of the global minima on the manifold cannot generalize.)
--- the results seem very intuitive and are a straightforward application of certain topological theorem.